# Benefits of realist evaluation for rapidly changing health service delivery

Justin Jagosh  ,[1] Hannah Stott,[1] Serena Halls,[1] Rachel Thomas,[1] Cathy Liddiard,[1] Margaret Cupples,[2] Fiona Cramp,[1] Paula Kersten,[3] Dave Foster,[1] Nicola E Walsh[1]

¹Faculty of Health and Applied Sciences, Glenside Campus, University of the West of England, Bristol, UK
²Department of General Practice, Queens University, Belfast, UK
³School of Health Sciences, Canterbury Christ Church University, Canterbury, UK

**Correspondence to**
Dr Justin Jagosh;
j.jagosh@realistmethodology-cares.org

## ABSTRACT

Realist evaluation is a methodology that addresses the questions: 'what works, for whom, in which circumstances, and how?'. In this approach, programme theories are developed and tested against available evidence. However, when complex interventions are implemented in rapidly changing environments, there are many unpredictable forces that determine the programme's scope and architecture, as well as resultant outcome. These forces can be theorised, in real time, and included in realist evaluation outputs for current and future optimisation of programmes. Reflecting on a realist evaluation of first-contact physiotherapy in primary care (the FRONTIER Study), five important considerations are described for improving the quality of realist evaluation outputs when studying rapidly changing health service delivery. These are: (1) ensuring that initial programme theories are developed through creative thinking sessions, empirical and non-empirical literature, and stakeholder consultation; (2) testing the causal impact of formal and informal (eg, emergent) components of service delivery models; (3) contrasting initial programme theories with rival theory statements; (4) envisioning broad system impacts beyond the immediate implementation setting; and (5) incorporating rapidly evolving service developments and context changes into the theory testing process in real-time (eg, Additional Role Reimbursement Scheme, COVID-19). Through the reflections presented, the aim is to clarify the benefit of realist evaluation to assess emerging models of care and rapidly changing health service delivery.

## INTRODUCTION

Realist methodology is used for investigating how programmes work, for whom and in which circumstances.[1 2] Although gaining widespread recognition for its value in assessing complexity across health and care sectors,[3] guidance is sparse on using the realist approach to assess evolving models of health service delivery embedded in rapidly changing contexts. Shifting clinical need, heavy caseloads, competing health policies and resource scarcity are a few reasons why health services may shift and evolve. Initiatives may also change due to practical necessities realised only during the implementation process. Complicating the evaluation of new efforts is the layering of interventions on existing services to address deficits and demands. As trends, needs and priorities shift, new models of service delivery may be terminated before substantial evidence of success or failure can accumulate. The resources of terminated interventions are often repurposed for newer initiatives, which are then launched again without a strong or insightful base of evidence. Realist evaluation is a suitable methodology to assess interventions under these conditions as the approach brings attention to the key hypothesised mechanisms of programmes and aspects of the context that matter, and such insights can be taken forward to other studies as programmes evolve and are repurposed. In this way, realist evaluations cumulate efforts so as to not 're-invent the wheel' with every study and can allow for learning from the relevant programme theorising that has come before.[1]

This paper provides reflections on the FRONTIER study, which is a UK-based *National Institute for Health Research* (NIHR) funded study (https://fundingawards.nihr.ac.uk/award/16/116/03) examining first-contact physiotherapy (FCP). FCP introduces specialist, and in some cases, advanced practice physiotherapists into primary care settings to assess, diagnose, treat and manage patients presenting with musculoskeletal (MSK) disorders without the requirement for a prior general practitioner (GP) consultation.[4] FCP is a model developed as an alternative to traditional primary care service for MSK management in which patients are seen by a physiotherapist only after receiving a referral from a GP. In the new model, FCPs work at primary care sites, allowing patients direct access through self-referral or reception triage. The FCP embedding process is complex and impacts GPs, reception staff, patients, physiotherapists and the wider health system. While FCP services have been in existence over the last decade, their contribution to the primary care workforce has significantly increased more recently in response to the National Health Service (NHS) Long

> **Box 1    Five important considerations for conducting a realist evaluation for rapidly changing health service delivery models**
>
> 1. Ensuring that initial programme theories are developed through creative thinking sessions, empirical and non-empirical literature, and stakeholder consultation.
> 2. Testing the causal impact of formal and informal (eg, emergent) components of service delivery models.
> 3. Contrasting initial programme theories with rival theory statements (ie, how the same resources can trigger very different responses and outcomes).
> 4. Envisioning broad system impacts beyond the immediate implementation setting.
> 5. Incorporating rapidly evolving service developments and context changes into the theory testing process in real-time (eg, Additional Role Reimbursement Scheme, COVID-19).

Term Plan (https://www.longtermplan.nhs.uk/publication/nhs-long-term-plan/) and actioned through the Additional Roles Reimbursement Scheme (https://www.england.nhs.uk/wp-content/uploads/2019/12/network-contract-des-additional-roles-reimbursement-scheme-guidance-december2019.pdf). In England, the aim is for all adults to have access to a FCP by 2024 (https://www.england.nhs.uk/gp/expanding-our-workforce/first-contact-physiotherapists/).

Five important insights for conducting a realist evaluation in the context of rapidly evolving contexts are presented in box 1. These considerations are explored further in the sections below, along with a brief description of the process of conducting a realist evaluation.

### Understanding realist evaluation

Realist evaluations of service delivery models typically involve the development of initial programme theories (IPTs), which are causal statements (eg, 'if…then') hypothesising *how* programme outcomes are manifested through programme mechanisms and corresponding contexts (see box 2 for an example). Protocols are developed to collect and analyse data to test the IPTs. Deductive (theory-testing) as well as inductive (theory-gleaning) activities are used to build a proposed process to uncover underpinning explanatory mechanisms. This is known as *retroduction* (see definition in box 2).[5 6] Realist evaluation uses context–mechanism–outcome configurations to achieve explanatory insights in theory development and data analysis. Programme mechanisms are understood to be underpinning generative forces that produce outcomes that activate in conducive contexts. Specifically, mechanisms are defined as the reasoning, response or reaction by stakeholders (eg, patients and staff) to programme resources. Programme resources may be formal or informal.[2 7] Context may include aspects of causal impact that reside outside the scope of the programme's architecture.[8 9] Outcomes can be quantitative or qualitative data typically seen as measurable impacts in behavioural, clinical or system-level terms (see box 2 for an example). Realist evaluations are often accompanied by a realist review (also known as realist synthesis),[10] which is a literature-based realist analysis of programme theories related to the interventions under scrutiny.

> **Box 2    Definition and examples of terms used in realist evaluation**
>
> **CONTEXT–MECHANISM–OUTCOME CONFIGURATION:** the central heuristic used in realist evaluation to understand what works, for whom, under which circumstances and how. Context is the backdrop of programmes, whereas mechanism is how stakeholders respond to resources. Outcome is measurable impact at the behavioural, clinical or system level. For example, *context: physiotherapists often have more specialist knowledge regarding musculoskeletal (MSK) conditions compared with general practitioners (GPs). Mechanism: first-point-of-contact physiotherapists are able to diagnose complicated MSK conditions in primary care and provide immediate access to tailored interventions for patients (resource) which reassures patients and physicians that patients are getting the timely MSK management they need (response). Outcome: improved patient outcomes and satisfaction; increased staff satisfaction; fewer appointments required in onward referral; upskilling of GPs; and fewer prescriptions.*
>
> **INITIAL PROGRAMME THEORY (IPT):** a hypothetical statement, often in the form of 'if…then,' that is developed at the start of a realist evaluation to explain how a programme or programme component works to produce outcomes. For example: '*If the primary care practice expects FCPs to allot a maximum of ten minutes to manage a MSK patient appointment in line with standard GP appointment length, and FCPs challenge this expectation based on a 20-minute min appointment length which is standard to traditional physiotherapy, then GPs may perceive FCPs as being inefficient and may look to employ other practitioners in the future*'.
>
> **RIVAL THEORY:** a hypothetical statement that shows how the same programme resources can lead to very different responses and outcomes. For example: *Rival theory A: physiotherapists working in primary care unburden GPs by attending to patients with MSK disorders. The reduced exposure to patients with MSK conditions results in GPs experiencing a deskilling of MSK expertise. Rival theory B: physiotherapists working in primary care expose GPs to expert MSK management, resulting in GPs upskilling their MSK expertise.*
>
> **CONTRASTIVE THEORY:** a hypothetical statement that explains how a programme strategy works in comparison with a different programme strategy: For example: *if a first-contact physiotherapist (FCP) does not have injecting or prescribing qualifications then they will rely on traditional physiotherapy modalities for patient care (ie, exercise, education and lifestyle approaches). This contrasts with FCPs who do have said qualifications and may use injections and prescriptions more readily. The FCP who uses traditional modalities may provide improved patient outcomes over time over those who do not, due to the holistic approach inherent to those modalities.*
>
> **PROGRAMME ARCHITECTURE:** the complete set of strategies/components that comprise an intervention, both formally allocated and advised as well as informally assembled and adapted from local resources and deficits. For example, formal architecture of FCP includes banding, appointment length, reception staff triage training and IT system integration. Informal architecture includes patient explanation about the FCP role, staff attitudes and spontaneous interprofessional coordination efforts between GPs and physiotherapists. Some aspects of the informal architecture may become formalised over time. Realist evaluation uncovers the mechanisms underpinning the programme's architecture.
>
> **RETRODUCTION:** a mode of inference that examines empirical outcomes in relation to the corresponding mechanisms of action that serve to produce them. For example: if a patient with an MSK disorder improves their condition by adhering to physiotherapy advice, it can be theorised that trust in the physiotherapist may increase patient motivation to uptake such advice. MSK improvement is the outcome, whereas trust and motivation are mechanisms.

## Five Important considerations for conducting a realist evaluation for rapidly changing health service delivery models

### Ensuring that initial programme theories are developed through a combination of creative thinking sessions, empirical and non-empirical literature, and stakeholder consultation

High-quality empirical literature is often sought to inform the development of IPTs at the outset of a realist evaluation. However, new service delivery models typically lack a trail of historical evidence regarding success and failure, and a lack of clarity on the programme's scope and architecture. These unknowns preclude easy identification of IPTs from pre-existing literature. For this reason, IPT development requires team-based creative thinking, consultation with key stakeholders at the early stages and retrieval of empirical as well as non-empirical literature. Such activities will bring initial programme theorising in line with current developments and will likely produce relevant hypothetical insights about the mechanisms at play, which may be obscured in literature sources. During literature review, data sources should be expanded to include unpublished (grey) literature, policy documents, online blogs and forums, and from professional body documentation.

For the FRONTIER study, a 6-month realist synthesis was conducted at the outset to develop the IPTs and establish a suitable research scope to examine the most important facets of the emerging FCP service delivery model. The body of high-quality empirical literature on FCP was found to be limited, and the few research papers retrieved did not describe clear programme theories that could be imported to the study. However, this literature was helpful to stimulate creative thinking and discussion within the team, which informed the development of relevant IPTs. Additional beneficial literature sources included policy documentation, online blogs and editorials in the UK-based physiotherapy professional body magazine 'Frontline'. In reading policy documentation, the description and scope of FCP provided insight into the programme's formal, expected architecture. In the blog and editorial literature, physiotherapists candidly described experiences, successes and concerns while working in general practice. Such sources of literature provided greater insight into the informal, unexpected aspects of the programme and contemporary implementation issues.

### Testing the causal impact of formal and informal (eg, emergent) components of service delivery models

IPTs developed and tested in a realist evaluation can account for both the formal and informal architecture of programmes (see box 2). This is particularly important for evolving models of health service delivery because as formalised resources are shifted or removed, informal efforts are often needed to keep programmes afloat. Although policy documentation on service reorganisation will describe formal architecture, implementation processes require additional undocumented efforts. A realist evaluation can capture these efforts, thus forming a comprehensive picture of how the programme works and how it evolves in real time.

For the FRONTIER study, the issue of pay scale, banding and associated skill-level exemplifies the importance of understanding formal and informal aspects of the programme. Documents from Health Education England (HEE) were scrutinised to understand the recommendations for how FCP roles should be implemented in primary care settings. HEE states that '*a First Contact Practitioner (FCP) is a diagnostic clinician working in Primary Care at the top of their clinical scope of practice at masters level Agenda for Change Band 7 or equivalent and above. This allows the FCP to be able to assess and manage undifferentiated and undiagnosed MSK presentations*'.[11] However, banding for FCP roles varies due to availability of appropriately banded staff for work in general practice. Given this reality, it was observed that the architecture of FCP changes depending on the banding of the physiotherapist recruited to a primary care team. Physiotherapists at higher bands, while having greater experience and clinical capability, were able to offer prescriptions and injection, which were frequently not within the scope of practice for lower banded physiotherapists working as FCPs. Through the realist evaluation it was theorised that, paradoxically, higher banded physios are potentially vulnerable to time pressures in primary care (ie, reducing appointment length) leading to increased prescribing and injecting as opposed to engaging patients with core physiotherapy interventions. Alternatively, it was theorised that physiotherapists who were not qualified to prescribe and inject were also not vulnerable to pressure to reduce appointment length. Traditional physiotherapy requires a longer appointment length than a GP appointment to engage patients with physical exercise, education and communication. Longer appointment length also means increased time to establish trust and rapport with the patient. While formal guidance outlines general principles of FCP banding and pay structure, variation in banding at the local sites means that primary care managers are left on their own to try to understand and make decisions regarding FCP service architecture. Realist evaluation can be used to theorise these complexities, and data collection can be conducted to better understand programme functioning given resource differences and limitations across local sites.

### Contrasting initial programme theories with rival theory statements (ie, how the same resources can trigger very different responses and outcomes)

Developing rival theories during the development of the IPTs can help to clarify aspects of the programme that are not well understood, especially in times of rapid context change. This is because as contexts change, alternative theories may explain resultant shifts in the programme's architecture as well as successes and failures. Rival theories hypothesise how the same programme resources can lead to very different mechanism responses and outcomes, given general expectations of a new initiative. Similarly, contrastive theories can show how resources of a new initiative are expected to work differently when compared with older established practices (eg, FCP vs GP – first models of care). Contrastive theories are also important to determine the clinical and cost-effectiveness of new efforts. Two advantages in including realist contrastive and rival theories are: (A) to help determine if new

initiatives layered on existing services yield at least no worse clinical and cost outcomes than standard practices alone; and (B) to contrast models to unearth elements of context which would otherwise remain obscured.

In the FRONTIER study, the inclusion of rival and contrastive ideas in theorising was valuable as it helped to explore the overburdened traditional model of GP-led MSK care delivery. The realist evaluation was used to understand how different GP practices contract and employ physiotherapists to tease out the causal contribution of different facets of the FCP model. One rival theory pertained to the upskilling and deskilling of GPs when physiotherapists are included in primary care. It was theorised that the presence of physiotherapists in primary care unburdens GPs by attending to many of the patients with MSK disorders but results in a deskilling effect for GPs in relation to their MSK expertise. A rival theory was also explored: that physiotherapists working in primary care help GPs upskill in relation to MSK issues by exposure to specialised knowledge, skills and innovations that the physiotherapist brings to the practice. These theories were then tested through data collection.

### Envisioning broad system impacts beyond the immediate implementation setting

New models of service delivery can mean moving resources from one part of a system to another (eg, moving staff from secondary to primary care). This movement of resources may relieve pressure in the destination area but create new staff challenges in the areas where resources originated. Such reorganisation efforts require that realist evaluators theorise the broader impacts of the initiative, as well as the longer term ripple effects. Although it may be difficult to capture evidence to test theories of broad and longer term impact, such theorising may still be useful for future research in the field, as well as for ongoing programme monitoring.

The FRONTIER study team explored the idea that FCP is a model that attracts physiotherapists from secondary to primary care settings. It is suspected that some cases this migration has resulted in the depletion of senior physiotherapists in secondary care settings with consequent impact on supervision of junior staff and waiting list times. Such resource depletion may call into question the overall benefits of the FCP initiative across the pathway and create new pressures on physiotherapy services that modify the FCP model of care. The realist evaluation can theorise such impacts and either collect data to test such theories or produce recommendations for future research to investigate the wider impacts.

### Incorporating rapidly evolving service developments and context changes into the theory testing process in real-time (eg, additional role reimbursement scheme, COVID-19)

Rapid shifts in service development may require realist evaluators to abandon theories developed at the outset of a study in favour of theories that become increasingly relevant over time. The abrupt rupture in the health service landscape brought about by the 2020 COVID-19 pandemic exemplifies the need to study causal impact of contemporary and emergent changes in real time. For rapidly changing models of service delivery, the output of the realist evaluation needs to account for new and divergent approaches that have emerged out of necessity over the course of the research.

The FRONTIER study collected data during the COVID-19 pandemic, which demonstrated how the pandemic dramatically impacted the FCP service delivery model. The UK-wide lockdown initiated in March 2020, alongside advice to stop the delivery of all non-essential face-to-face health and social care[12] created conditions for new programme theorising in the realist evaluation. For the physiotherapy profession, the acceleration of online consultations meant an immediate and considerable change to physiotherapy provision with a rapid shift in the way consultations were conducted and managed. Although this service change was planned for future implementation (https://www.longtermplan.nhs.uk/online-version), the advent of the pandemic expedited its introduction and created little time for preparation or training. Realist evaluation can be used to theorise and investigate emerging developments by consulting expert practitioners and reviewing current policy documents. During FRONTIER data collection, it was found that while some FCP practitioners considered the transition to online service provision to be a beneficial service development that saved time and was more convenient for many patients, others questioned whether patients considered it a 'valid' consultation and raised concerns regarding the greater potential for misdiagnosis and service inefficiencies. These rival theories served to improve the vision of new service architecture when the current evidence-base was lacking. From a longitudinal perspective, there is a potential for the shift to online service provision to have long-term implications for physiotherapy practice and therefore FCP implementation, although the extent of this impact remains uncertain. In addition to the impact on practice, the shift has had implications for the proposed FCP programme theories, as the original IPTs for the FRONTIER study were based on physical ('in house') colocation of FCPs with GPs in primary care. It is possible in a realist evaluation to construct new IPTs during data collection and test those theories with the remaining resources of the study.

## DISCUSSION

There is a need to maintain an iterative and adaptive position to theorising in realist evaluation when contexts change rapidly, or the architecture of programming remains unknown. It may also feel necessary to abandon early theorising if this becomes out of step with developments. These are normal processes in realist evaluation, and adequate time should be given to achieve clarity regarding important aspects of programmes for theory testing. The rapid evolution of new initiatives can be due to implementation barriers and resource constraints and the timeframe of a realist evaluation may overlap on such rapid change. In this regard, the foci of

theory development and testing may shift over the duration of a realist investigation. This is advantageous as the outputs of such realist evaluations may yield important insights regarding programme success and failure that can be carried forward in future research and programme monitoring. In addition, the creative thinking that occurs at the outset of a realist study can be beneficial to study programmes in evolution, as this increases the agility that research teams need to adapt theory and research protocols. Inevitably, some theories will always be difficult to test due to a lack of available data. Nonetheless, the development of such theories contributes to a cumulative body of work that lends itself to future study. Initial theorising may also yield an overabundance of theories (eg, n>30), requiring teams to consolidate and prioritise those theories for testing. An important lesson from the FRONTIER study is that the creative thinking to develop IPTs conducted at the outset of the study was invaluable even if not all those theories were taken forward in data collection. It was found that the initial stage of the realist investigation was as much 'theory development' as it was 'theory sensitization'.

As health systems evolve worldwide, it is necessary that methodologies such as realist evaluation are used and developed to capture the real-time changes and corresponding causal impacts to serve the needs of programme implementation. Through the reflections presented in this paper, we hope our demonstration of realist evaluation and experience in the FRONTIER study will help other teams improve the design of studies used to assess emerging and rapidly changing service delivery models.

**Contributors** JJ was the primary writer and was responsible for submitting the manuscript. Coauthors: HS, SH, RT, CL, MC, FC, PK, DF and NEW provided multiple rounds of feedback. All authors were involved in the design and analysis phases of the FRONTIER project.

**Funding** The FRONTIER study is funded by the National Institute for Health Research (NIHR) (HS&DR) (16/116/03)/HS&DR).

**Disclaimer** The views expressed are those of the authors and not necessarily those of the NIHR or the Department of Health and Social Care. No additional funding was allocated for the development of this manuscript.

**Competing interests** None declared.

**Ethics approval** Not applicable.

**Provenance and peer review** Not commissioned; externally peer reviewed.

**ORCID iD**
Justin Jagosh http://orcid.org/0000-0001-6807-2957

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
