## [Reviewer comments · BMJ Open]

ARTICLE DETAILS

TITLE (PROVISIONAL)	The benefits of realist evaluation for rapidly changing health service delivery models
AUTHORS	Jagosh, Justin; Stott, Hannah; Halls, Serena; Thomas, Rachel; Liddiard, Cathy; Cupples, Margaret; Cramp, Fiona; Kersten, Paula; Foster, Dave; Walsh, Nicola

VERSION 1 – REVIEW

REVIEWER	Bethune, Rob Royal Devon and Exeter NHS Foundation Trust
REVIEW RETURNED	23-Mar-2022

GENERAL COMMENTS	Thank you for allowing me to review this interesting and relevant article. As BMJ Open is not a solely qualitative journal, I think some readers will struggle with the more detailed qualitative language and terminology. I am not saying that it should be 'dumbed' down, I just think that it will be too esoteric for much of the readership of BMJ Open. Although a clinician, I have been part of several qualitative research projects, including using the realist approach, and I found the wording, in parts, difficult to follow. (examples:- 'counter-factual', 'retroductive' programme 'architecture' 'realist contrastive and rival theories') Very interesting point regarding the benefit on online blogs over more mainstream empirical work. The sections 'Lessons from FRONTIER:' are much easier to read and really help with understanding the arguments. There is nothing wrong with the argument and the findings, but it could be made easier to read. The discussion is excellent and much easier to read than the rest of the paper. I hope these comments help.
--

REVIEWER	Power, Jessica University of Dublin Trinity College, Centre for Global Health
REVIEW RETURNED	05-May-2022

GENERAL COMMENTS	Thank you for your contribution. These reflections are of value to future researchers when undertaking RE in an evolving context.
---

VERSION 1 – AUTHOR RESPONSE

Reviewer: 1

Dr. Rob Bethune, Royal Devon and Exeter NHS Foundation Trust Comments to the Author:

Thank you for allowing me to review this interesting and relevant article.

6. As BMJ Open is not a solely qualitative journal, I think some readers will struggle with the more detailed qualitative language and terminology. I am not saying that it should be 'dumbed' down, I just think that it will be too esoteric for much of the readership of BMJ Open. Although a clinician, I have been part of several qualitative research projects, including using the realist approach, and I found the wording, in parts, difficult to follow. (examples:- 'counter-factual', 'retroductive' programme 'architecture' 'realist contrastive and rival theories')

This is a very helpful comment and to increase the accessibility of the manuscript, we have simplified terminology throughout the manuscript and have re-worded many parts of the manuscript to bring increased clarity to the ideas presented. We have also added a table presenting a definition of terms. The table includes general definitions for realist terminology as well as examples to clearly illustrate the points made. These terms are key for realist methodologists, and we anticipate the manuscript to become highly referenced due to the definitions presented.

This addition has somewhat increased the word count of the manuscript, however we believe the added table adds considerable value, and allows the manuscript to be much more accessible to a wider audience than it was before.

VERSION 2 – REVIEW

REVIEWER	Bethune, Rob Royal Devon and Exeter NHS Foundation Trust
REVIEW RETURNED	05-Jul-2022
GENERAL COMMENTS	Thank you for asking me to re-review this paper. I now think it is excellent and ready for publication. I particularly like the table explaining the different terms in realist/qualitative evaluation. BW Rob Bethune